

# Updated observations of clouds by MODIS for global model assessment

Robert Pincus[1], Paul A. Hubanks[2,3], Steven A. Platnick[3], Kerry Meyer[3], Robert E. Holz[4], Denis Botambekov[4], and Casey J Wall[5]

[1]Lamont-Doherty Earth Observatory, Columbia University, Palisades, New York, USA
[2]Adnet Systems, Inc, Bethesda, Maryland, USA
[3]Earth Science Division, NASA Goddard Space Flight Center, Greenbelt, Maryland, USA
[4]Cooperative Institute for Meteorological Satellite Studies, University of Wisconsin, Madison, Wisconsin, USA
[5]Scripps Institution of Oceanography, University of California San Diego, San Diego, California, USA

**Correspondence:** Robert Pincus (Robert.Pincus@columbia.edu)

**Abstract.** This paper describes a new global dataset of cloud properties observed by MODIS relying on the current (collection 6.1) processing of MODIS data and produced to facilitate comparison with results from the MODIS observational proxy used in climate models. The dataset merges observations from the two MODIS instruments into a single netCDF file. Statistics (mean, standard deviation, number of observations) are accumulated over daily and monthly time scales on an equal-
angle grid for viewing and illumination geometry, cloud detection, cloud-top pressure, and cloud properties (optical thickness, effective particle size, water path) partitioned by thermodynamic phase and an assessment as to whether the underlying observations come from fully- or partly-cloudy pixels. Similarly partitioned joint histograms are available for (1) optical thickness and cloud-top pressure, (2) optical thickness and particle size, and (3) cloud water path and particle size. Differences with standard data products, caveats for data use, and guidelines for comparison to the MODIS simulator are described. Data
are available on daily (NASA, 2022a, doi:10.5067/MODIS/MCD06COSP_D3_MODIS.062) and monthly (NASA, 2022b, doi:10.5067/MODIS/MCD06COSP_M3_MODIS.062) timescales.

## 1 MODIS observations of clouds and the assessment of global models

The distribution of cloud radiative properties strongly impacts Earth's energy balance (e.g. Hartmann et al., 1992). Uncertainty in how clouds will respond to anthropogenic forcing is responsible for most of the difficulty in estimating climate sensitivity
(Sherwood et al., 2020), largely because clouds are so tightly coupled to atmospheric circulations (Bony et al., 2015) across a wide range of scales. Clouds' impacts on the global and local energy budgets, and on the distribution of precipitation, motivate efforts to assess the distribution of clouds produced by global models against observations (e.g. Pincus et al., 2012; Klein et al., 2013).

Observations from space offer the most globally uniform observations of clouds but direct comparisons with predictions from
numerical models are complicated by differing definitions of cloudiness, sampling errors introduced by the observing system, and differing scales in the observations as compared to the models. The "ISCCP simulator" (Yu et al., 1996; Klein and Jakob,



1999; Webb et al., 2001) was introduced several decades ago to allow for more informative comparisons. The ISCCP proxy is software that runs within a climate model and roughly maps the clouds as represented by the model to synthetic observations as would be obtained from the ISCCP program (Rossow and Schiffer, 1991). Insights from the ISCCP simulator inspired a range

of other such proxies focused on clouds, many of which have been packaged together in the CFMIP Observation Simulator Package (COSP, see Bodas-Salcedo et al., 2011; Swales et al., 2018).

Among other simulators COSP includes a proxy for MODIS observations of clouds as described in Pincus et al. (2012). Compared to ISCCP and other passive sensors MODIS, described more fully below, offers better characterization of cloud thermodynamic phase and routine estimates of cloud particle size. Synthetic observations from the MODIS simulator, namely

joint probability distributions of cloud optical thickness and particle size, are requested as part of phase 3 of the Cloud Feedbacks Model Intercomparison Project (Webb et al., 2017). Output is requested only for the daylit portion of the globe, where richer information is available from passive sensors.

Because the MODIS proxy is most widely used alongside other proxies within COSP the output is normally configured to complement the other proxies in the suite. A number of barriers arise in comparing this focused subset of data with the standard

observational datasets produced by the MODIS Science Team (e.g. King et al., 2013) ranging from mundane but important hurdles, such as the files being in different formats, to more fundamental issues such as data produced in the simulator having no direct counterpart in the observational data. Pincus et al. (2012) described a dataset designed to lower those barriers. That dataset, produced with an earlier collection of the underlying MODIS data, post-processed standard monthly aggregations and invoked a number of assumptions to make the observations more compatible with fields produced by the simulator. The system

was also quite fragile and ceased production when NASA updated the production of MODIS datasets. Observations up to September 2016 remain available at https://ladsweb.modaps.eosdis.nasa.gov/archive/NetCDF/L3_Monthly/V02/.

Here we describe a new global dataset of cloud properties observed by MODIS relying on the current processing of MODIS data and produced to facilitate comparison with results from the MODIS simulator. The new data, designated MCD06COSP, combines MODIS pixel-scale observations of cloud occurrence, cloud top pressure, and cloud optical properties from Terra

(MOD06_L2) and Aqua (MYD06_L2) on daily and monthly timescales. The dataset, produced using a system designed to be more robust to changes in the upstream data, provides a set of custom cloud-related parameters using specific dataset definitions more closely aligned with the MODIS simulator than are the standard datasets. Data are provided in the Network Common Data Format Version 4 (NetCDF4) format that is widely used to distribute climate model data.

This paper documents the MODIS COSP Level-3 dataset with emphasis on helping users interpret the observations and

make informed comparisons to results from the MODIS simulator. This document summaries and expands upon a longer and more complete Users' Guide available at https://atmosphere-imager.gsfc.nasa.gov/products/monthly_cosp/documentation.

## 2 How data are produced

MODIS, the Moderate Resolution Imaging Spectroradiometer (Salomonson et al., 1989), is a 36-channel narrowband imaging instrument developed for NASA's Earth Observing System. Two MODIS instruments were launched near the beginning of



the 21st century and continue to provide data as of this writing. NASA's MODIS science team produces a wide range of observational datasets based on measurements from the sensor including the cloud-related observations described below.

The discussion that follows adopts the MODIS project's terminology to describe in detail how the data are produced. In this terminology, observations at the native resolution (250 m to 1 km for the MODIS instrument) are referred to as pixels, which are acquired and processed in five-minute granules corresponding to 2030 1 km pixels along the satellite track by 1354

pixels cross-track (nominally 2330 km, because pixels sizes increase with scan angle). Each version of the software used in the data processing stream is referred to as a Collection. At the time of this writing data are produced using Collection 6.1 as documented in Baum et al. (2012) and Platnick et al. (2017). Data are produced at three distinct levels: Level-1 refers to calibrated geo-located radiances (near-raw data); Level-2 describes retrievals (inferences) of geophysical and/or optical quantities at the pixel scale; Level-3 means observations aggregated in space and time, including the dataset described here.

## 2.1 Identifying clouds and determining their properties

Here we briefly describe how Level-2 data are produced with the intent of orienting users to the data being provided. For further details on the production of pixel-scale observations see Platnick et al. (2017), Baum et al. (2012) as well as Pincus et al. (2012) and references therein.

Pixel-scale (Level-2) estimates of cloud properties are determined in two steps: the first determines the likelihood that a

given pixel contains clouds; the second estimates cloud properties for cloudy pixels. (A separate processing step determines aerosol properties in the non-cloudy pixels.) Cloud detection relies on a decision tree involving multiple channels and produces a cloud mask at 1 km resolution. The decision trees use different information whether the pixel is sunlit ("daytime") or not using the criterion that the solar zenith angle is less than $85°$. Each pixel is flagged according to whether the cloud mask has been determined; determined pixels are flagged with one of four values (confidently cloudy, probably cloud, probably clear,

confidently clear). Cloud fraction at 5 km scale is determined as the ratio of confidently and probably cloudy pixels to all determined pixels. Cloud top pressure $p_c$ is determined at 5 km scale when at least four of the 25 1 km sub-pixels are cloudy (i.e. when 5 km cloud fraction equals or exceeds 16%). Cloud top pressure is estimated using $CO_2$ slicing (Menzel et al., 1983) for clouds with tops above about 700 hPa and thermal emission for lower clouds.

Cloud properties – cloud phase (liquid/ice), optical thickness $\tau_c$ and effective particle size $r_e$ – are estimated for sunlit

pixels flagged as confidently or probably cloudy. Cloud property retrievals define daytime slightly more conservatively (solar zenith angle less than $81.3731° = \arccos(.15)$) than does the cloud mask. Pixels identified by the mask as cloudy are excluded from retrievals if multi-spectral tests suggests that they are sunglint or heavy aerosol. Partly-cloudy (PCL) pixels are identified based either on their proximity to clear pixels ("cloud edges") or on the variability of the cloud mask at 250 m scales (Platnick et al., 2017). Cloud phase is determined with a weighted voting approach using a variety of spectral observations as well as

cloud retrievals (Marchant et al., 2016). Where measurements are ambiguous the pixel is labeled as such and liquid water is assumed in further calculations. Cloud optical thickness and particle size are estimated by minimizing the difference between two observations, one in a spectral channel in which condensed water absorbs and another in a channel in which liquid and ice scatter conservatively, and forward calculations at these wavelengths made as a function of $\tau_c$ and $r_e$ (as well as viewing





and illumination geometry, e.g. Nakajima and King, 1990). The thermodynamic phase determines the microphysical model,
and hence the particle shape and refractive index, used in the forward calculations. Three independent estimates are reported,
one each using observations at 1.6, 2.1, and 3.7 $\mu$m for the absorbing channel, in addition to a joint 1.6 and 2.1 $\mu$m retrieval
over snow/ice and open ocean surfaces. Absorption by condensed water increases with wavelength across these intervals, so
that the particle size estimated becomes increasingly representative of values near cloud top (Platnick, 2000), but estimates
using wavelengths at which condensate is more absorptive are less biased by sub-pixel variability (Zhang et al., 2012, 2017).
Retrievals using 3.7 $\mu$m are aggregated in the MODIS COSP Level-3 dataset.

## 2.2 Aggregating statistics in time and space

The MODIS COSP Level-3 dataset MCD06COSP is based on pixel-scale (Level-2) datasets which, at the time of this writing,
are produced using the Collection 6.1 data processing stream. NASA produces separate data for the MODIS instruments on
the Terra platform (1030 am LT nominal daytime equatorial crossing) and Aqua platform (part of the A-train constellation of
satellites described in e.g. L'Ecuyer and Jiang (2010), 1330 pm nominal daytime equatorial crossing). Terra was launched in
2000 and Aqua in 2002; because the MODIS COSP dataset includes observations from both platforms it is available only since
July 2002.

The spatial resolution of MODIS Level-2 data varies with observation type including the underlying resolution of the MODIS
instrument in the channels used to make the observation, with some fields available at nadir resolutions as fine as 250 m.
Geolocation information from the Level-2 cloud files is used for Level-3 aggregations, despite its availability at the relatively
coarse resolution of 5 km. Both the standard MODIS Level-3 data and the MODIS COSP data report statistics based on data
subsampled to the spatial resolution of the geolocation information. This choice, initially inspired by Sèze and Rossow (1991),
has a very small impact on mean values at 1° resolution (Oreopoulos, 2005). The precise sampling is adjusted to provide more
robust statistics. In particular, statistics are computed using the 1 km pixel that is one along-track position aft of the of the
center of each 5 km pixel (i.e. 1 km pixels 4 and 8 vs. 3 and 7). The change was originally motivated by a detector failure in
the 1.6 $\mu$m band in the Aqua instrument but has been adopted throughout the Terra and Aqua chain with no impacts on the
aggregated statistics (Oreopoulos, 2005).

Files are produced at daily and monthly time intervals. The number of observations available varies quite widely on daily
time scales, even after data from the Terra and Aqua platforms are combined (Figure 1). Both platforms have 16-day return
periods so that sampling through the month is quite zonally uniform, with increased sampling density poleward of 60° in the
summer hemisphere due to overlapping orbits and reduced sampling poleward of ∼53° in the winter hemisphere due to limited
illumination.

Starting with Collection 6, standard MODIS Level-3 products (King et al., 2013) use a somewhat involved decision tree
aimed at minimizing gaps, overlap, and the aggregation of data from two orbits almost 24 hours apart in a single file, when
deciding which pixels to include in each day's aggregation. Experience with the standard product, however, showed that
this decision introduced other artifacts in some fields (especially the cloud fraction from the cloud mask). In the interests of



Earth System
Science
Data

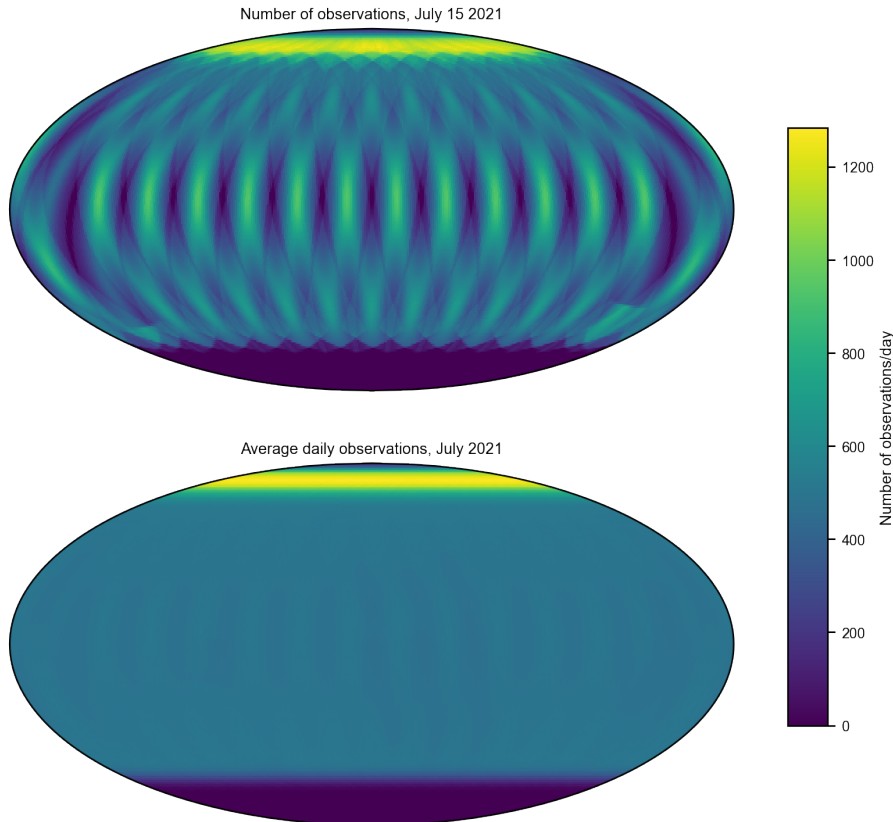

**Figure 1.** Number of daily observations in each $1°$ grid box in the MODIS COSP data product, which aggregates only daytime observations, on a single day (upper) and averaged over the course of a month (lower). Sampling density is illustrated with the number of observations used in determining the cloud mask fraction during July 2021.

simplicity the MODIS COSP product reverts to the practice in place before Collection 6: pixels are assigned to days according to the UTC time at the start of the granule's acquisition.

Consistent with standard MODIS datasets, two sets of cloud fraction estimates are produced, one based on results from cloud detection/masking and a second based on the phase and cloud optical properties determination. Total cloud fraction from the cloud mask is computed by averaging the cloud fraction computed at 5 km scale; as noted in the last section the 5 km cloud fraction is is ratio of the number of 1 km pixels flagged as probably or confidently cloudy to the number of pixels for which the cloud mask was determined, so that "undetermined pixels" are not included. Cloud fractions are also computed for high ($p_c < 440\,\text{hPa}$), middle, and low ($p_c \geq 440\,\text{hPa}$) clouds based on the determination of cloud-top pressure $p_c$. The total fraction

from the optical property retrieval step is the ratio of the number of pixels for which retrievals were successfully performed to the number of pixels for which the cloud mask was determined (including those pixels for which retrievals could not be performed). Estimates of the fraction of clouds identified as liquid and ice are also provided. The total retrieval cloud fraction



includes pixels for which the thermodynamic phase could not be determined and so may be larger than the sum of the liquid-and ice-cloud fractions.

Scalar measures of cloud optical properties for fully-cloudy pixels are aggregated in time and space. These properties include cloud optical thickness $\tau_c$ and its base 10 logarithm, particle size $r_e$, and the condensed water path estimated from the product of optical thickness and particle size. The logarithmic mean provides a useful estimate of time-mean reflectivity (Pincus et al., 2012). With the exception of the logarithmic mean, scalar measures are also reported separately for partly-cloudy pixels. Statistics represent the underlying population so that days with more observations, either because the sampling is denser (Fig.

1) or because clouds are more wide-spread, contribute more to the time-mean than do days with fewer observations. This approach differs from the standard MODIS Level-3 product in which monthly means of cloud fraction and cloud-top pressure are the linear average of daily means.

    The dataset also provides joint histograms summarizing the co-variability of cloud optical thickness with cloud particle size (computed separately for liquid and ice clouds) and with cloud top pressure (separated by phase, and for all clouds). Summing

the latter over all optical thickness bins and reducing the resolution in cloud-top pressure allows users to compute high, middle, and low cloud fractions consistent with cloud optical properties (as opposed to the cloud mask). Both sets of joint histograms are accumulated for both fully- and partly-cloudy pixels.

    Anticipating increasing interest in evaluation of aerosol-cloud interactions, the MODIS-COSP dataset also provides joint histograms of cloud water path and cloud particle size. Such histograms are not yet available from the MODIS simulator

within COSP though we anticipate adding such a diagnostic in the coming months. As with the cloud optical thickness joint histograms, results are reported separately for fully- and partly-cloud pixels.

### 2.2.1   Technical implementation

Although the MODIS COSP dataset MCD06COSP is produced by the MODIS Atmospheres Science Team, it is implemented in a separate data stream from the operational products. Level-2 data are staged to the Atmosphere Science Investigator-led

Processing System run by the University of Wisconsin Madison that supports continuity of data available between MODIS and follow-on sensors including Suomi National Polar-orbiting Partnership. Pixel-scale data are aggregated to Level-3 daily files, and daily files to monthly files, using custom software called "Yori" after a character in the 1982 movie "Tron". Yori accumulates statistics in time and space. Filtering and other transformations are accomplished by adding relevant fields to the Level-2 data. To compute cloud fractions, for example, a field with binary values – 1 where the pixel is considered cloudy by

the relevant definition and 0 where the pixel is not – is added to the Level-2 files; in the aggregated Level-3 file the mean of this field represents the cloud fraction. Cloud fractions resolved by height or phase require fields in which pixels are assigned 1 only when they are cloudy and conform to another condition.

    Yori computes mean and standard deviations for each variable. To enable aggregation over time that represent the underlying distributions, Yori also records the sum, sum-of-squares, and observation numbers for each variable. Yori is also able compute

the joint histogram of two variables. Joint histograms are accumulated as counts (pixels) in each bin and must be normalized before use for most applications.





## 2.3 Data contents and formatting

Data fields available in the MODIS COSP data are numerated in Tables 1 and 2. Data are produced on a rectangular latitude-longitude grid with 1° spatial resolution. One file combining measurements from both platforms is produced for each day; daily
files are accumulated into monthly files as described above. File names are constructed from a fixed prefix (MCD06COSP), a temporal resolution identifier (D3 or M3 denoting daily or monthly files respectively), the instrument name, the letter A and the acquisition date as a four-digit year and a three-digit day-of-year, the collection number, the production date and time, and a file format suffix. As an example, a daily file containing gridded observations from Collection 6.1 from a single day in July 2021 might be "MCD06COSP_D3_MODIS.A2021196.062.2022124015916.nc."

The files use netCDF4's ability to organize data in groups. Each netCDF name listed in Table 1 corresponds to one such group, each of which contains variables Mean, Sum, Sum_Squares, Standard_Deviation, and Pixel_Counts (see Sec. 2.2.1). Six groups also contain joint histograms, i.e. group Cloud_Optical_Thickness_Liquid contains the variable JHisto_vs_Cloud_Particle_Size_Liquid. optical thickness and cloud top pressure are discretized in 7 bins in these joint histograms, while particle size is discretized in 6 bins; the values defining histogram bins are recorded in attributes of the JHisto
variable. All variables are functions of latitude and longitude.

## 3 What to expect from the data

### 3.1 Caveats, known issues

An unidentified bug in cloud mask at Level 2 means a small number of pixels with valid data are ignored. This means the number of valid pixels is sometimes under-reported and can occasionally even be smaller than the the counts for retrievals.
As described above, joint histograms (Table 2) of optical thickness and cloud water path with particle size, and of optical thickness with cloud-top pressure, are reported as the number of pixels falling in each $(\tau, r_e)$, $(CWP, r_e)$, or $(\tau, p_c)$ interval. These may be converted to measures of cloud fraction in each bin by dividing the number of pixels by the number of observations that might have contributed, i.e. by the Pixel_Counts variable associated with the Cloud_Retrieval_Fraction_Total group.

Because data derived from the cloud mask use a different threshold for solar zenith angle than do fields derived from cloud retrievals, cloud fractions may differ markedly near the terminator (i.e. at the most polar latitudes for which data are available near the equinoxes) for small proportion ($< 0.5\%$) of grid cells.

### 3.2 Measures of cloudiness

#### 3.2.1 Mask and retrieval cloud fractions

As explained in Section 2.2 the MODIS COSP Level-3 dataset includes two estimates of cloudiness (cloud fraction), one computed from the cloud mask and one summarizing the frequency with which cloud properties have been estimated. The two

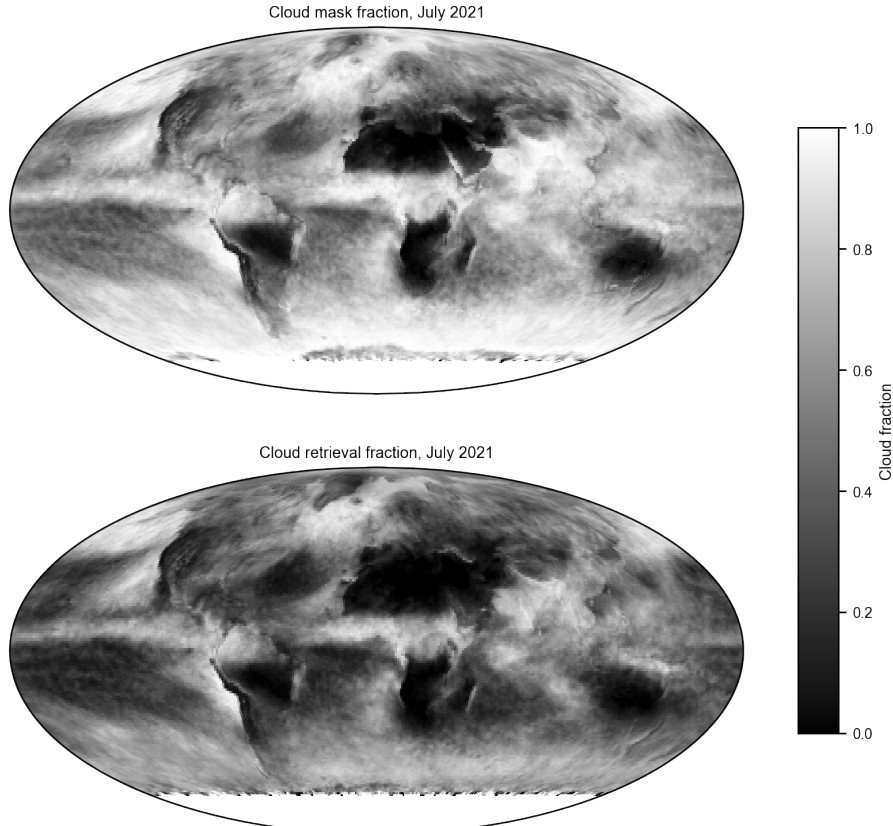

**Figure 2.** Cloud fraction as determined by the cloud mask (top) and the total determined from the retrievals (bottom) in July 2021. The cloud mask fraction is the ratio of those pixels for which clouds are probably or confidently detected to the number of pixels for which the cloud mask could be determined. The total cloud retrieval fraction excludes pixels identified as sunglint, heavy aerosol, or partly-cloudy. As a result cloud mask fraction is larger than cloud retrieval fraction almost everywhere, with exceptions only in areas near the winter time, high-latitude terminator.

steps have somewhat different aims: the cloud mask seeks to identify pixels unlikely to be clear sky (i.e., filtered for clear sky retrievals), while the retrieval step seeks to provide accurate estimates of cloud properties and so filters out pixels thought to provide unreliable estimates via "clear-sky restoral". The difference between the two estimates for an example month, July 2021, is shown in Figure 2). The removal of partly-cloudy pixels is responsible for much the roughly 22% difference: the top panel of Figure 3 shows the difference between cloud fraction estimates provided by the mask and the retrievals (i.e. the top and bottom panels of Fig. 2), respectively; the bottom panel shows that the difference is greatly reduced by accounting for the fraction of partly-cloud pixels.


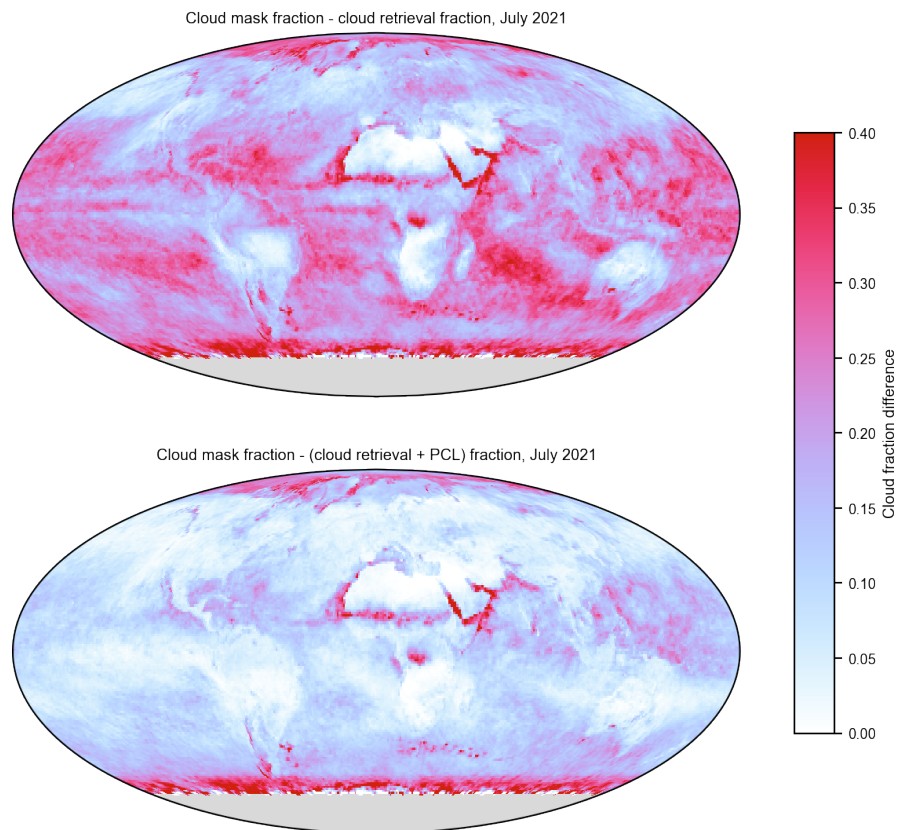

**Figure 3.** Difference between cloud fraction determined from the cloud mask and the cloud property retrievals neglecting (top) and including (bottom) pixels flagged as partly-cloudy (PCL) by the "clear-sky restoral" filtering step in July 2021. The fraction of partly-cloudy pixels is computed by summing over the associated joint histogram of cloud optical thickness and cloud top pressure (Table 2).

### 3.2.2 Vertical distribution of cloud (mask) fraction

The vertical distribution of total cloudiness, expressed as the proportion of high, middle, and low-topped clouds, is shown for July 2021 in Figure 4. Cloud top pressure is at 5 km resolution where the 1 km cloud mask fraction exceeds 16% (see Sect. 2.1), so the sum of the three height-resolved clouds fractions is, in some cases, slightly (less than 1%) smaller than the cloud fraction derived from the cloud mask.

### 3.2.3 (Retrieval) cloud fraction by phase

As described in Sect. 2.1 the determination of cloud thermodynamic phase is the first step in the retrieval of cloud optical properties and sets the microphysical model used in reporting these retrievals. Liquid clouds are substantially more common than are ice clouds (see Figure 5). Cloud phase is determined for more than 99% of all fully-cloudy pixels, i.e. the total retrieval

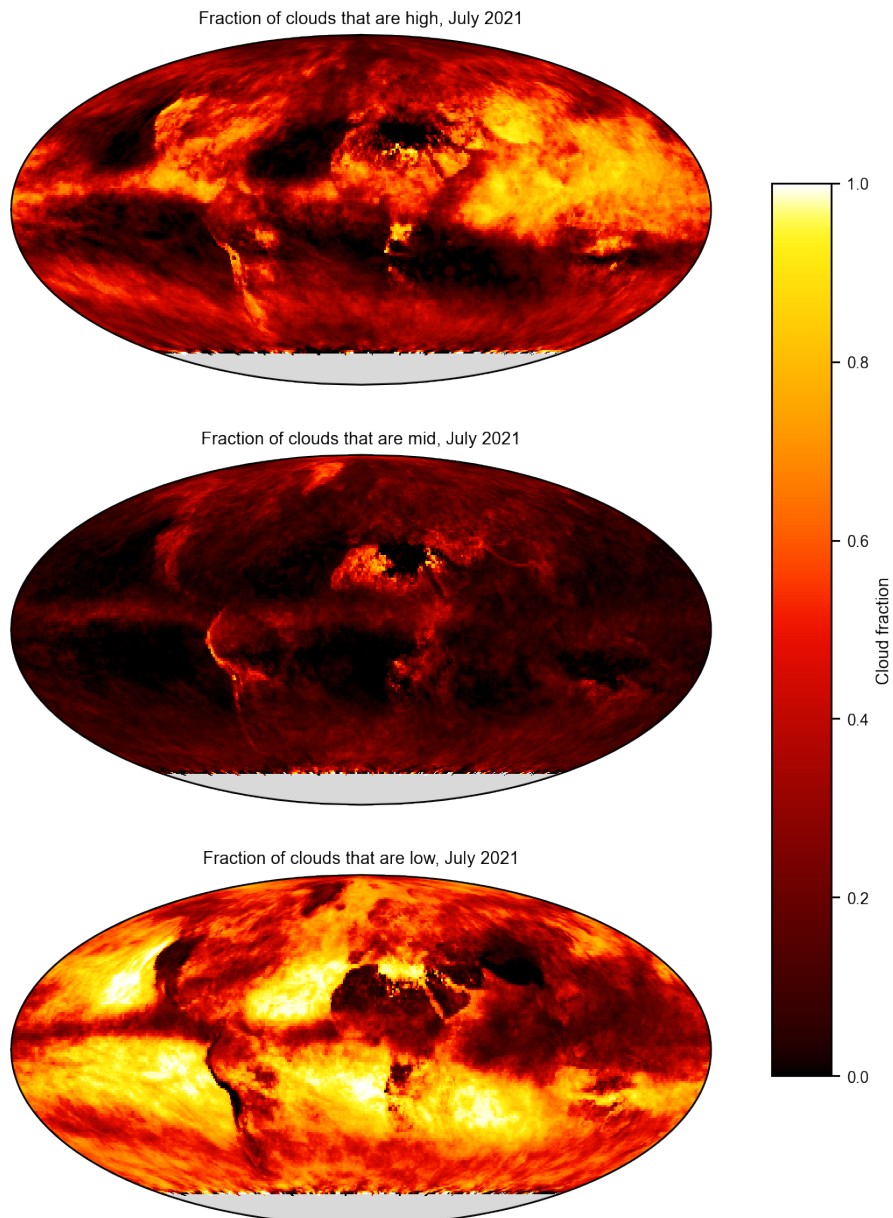

**Figure 4.** Proportion of clouds determined to be high (cloud top pressure $p_c < 440\,\mathrm{hPa}$), mid-level clouds ($680 < p_c \leq 440\,\mathrm{hPa}$) and low clouds($p_c \geq 440\,\mathrm{hPa}$) for the month of July 2021.

cloud fraction, which includes pixels for which the thermodynamic phase could not be determined, is larger than the sum of the liquid- and ice-cloud fractions by 0.9% in the global mean. Note that the retrieved phase is weighted toward upper cloud



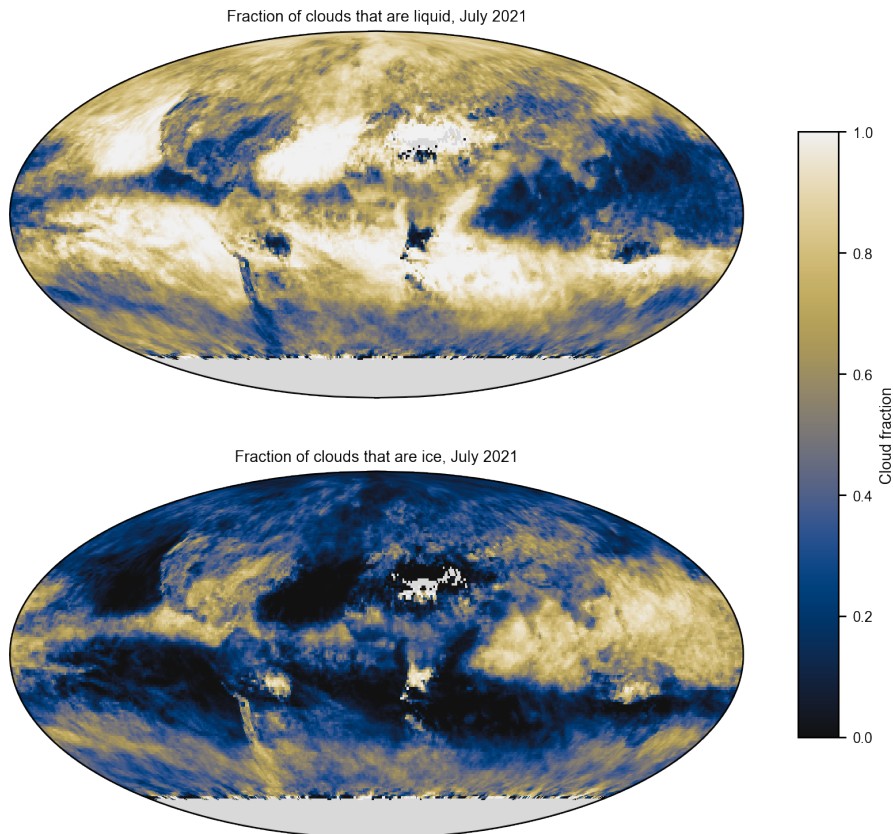

**Figure 5.** Proportion of clouds for which thermodynamic phases is determined to be liquid (top) and ice (bottom panel) in July 2021. MODIS determines the phase (and uses the relevant cloud microphysical model for most clouds: 94% of 1°grid cells have less than 5% of clouds with undetermined phase during this illustrative month.

layers when multilayer and multiphase clouds are present (an observational artifact treated by the simulator), i.e., ice cloud layers with significant optical depth overlying lower-level liquid clouds are retrieved as ice.

### 3.3 Time-averaging: optical thickness and albedo

Though optical thickness is the fundamentally retrieved quantity its linear average is of limited utility since both albedo (relevant for calculations of shortwave reflectivity) and emissivity (relevant for longwave calculations) depend non-linearly on

optical thickness. The albedo of a distribution of optical thickness is well-approximated by the albedo of the geometric mean optical thickness ($10^{\overline{\log_{10}\tau_c}}$) (Pincus et al., 2012). Figure 6 compares the arithmetic and geometric mean optical thickness for all clouds during July 2021; spatial variations in the latter are substantially smaller.

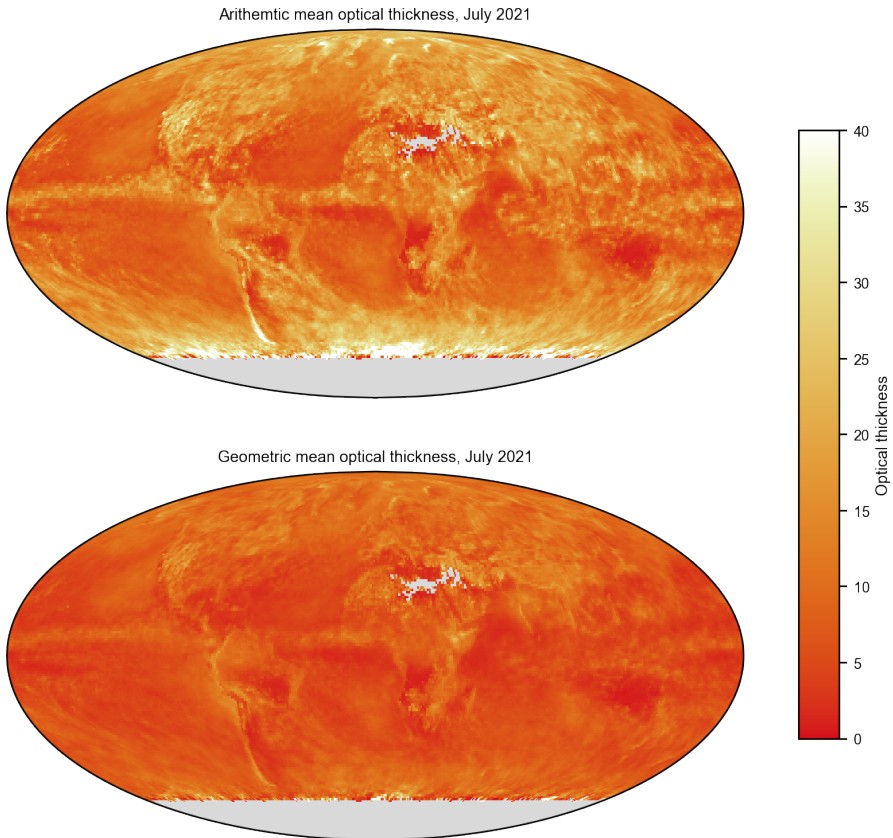

**Figure 6.** Arithmetic (top) and geometric (bottom) mean cloud optical thickness for all clouds in July 2021. Geometric mean optical thickness is a better predictor of time-average albedo and shows substantially less spatial variation.

## 3.4 Joint distributions

Both the MODIS and ISCCP simulators produce joint histograms of cloud optical thickness $\tau_c$ and cloud-top pressure $p_c$,
though the difference between the simulators is not as marked as the difference between the observations (Pincus et al., 2012), where the $CO_2$ slicing used by MODIS assigns more clouds to lower pressures than does the temperature-based characterization from ISCCP. The MODIS-COSP dataset further resolves the histograms by thermodynamic phase and for fully- and partly-cloudy pixels (Tab. 2). Global mean (area-weighted) histograms are shown in Figure 7. The vast majority of partly-cloudy pixels turn out to be liquid, low ($p_c > 800\,\mathrm{hPa}$), and optically thin ($\tau_c \le 3.6$), suggesting that the algorithms used to identify
these pixels are performing as designed.

Global mean joint histograms of $\tau_c$ with particle size $r_e$ are shown in Figure 8. Optical thickness is accumulated in the same bins as the $\tau_c$-$p_c$ histograms, which themselves follow ISCCP conventions; particle size bins are spaced unevenly to resolve




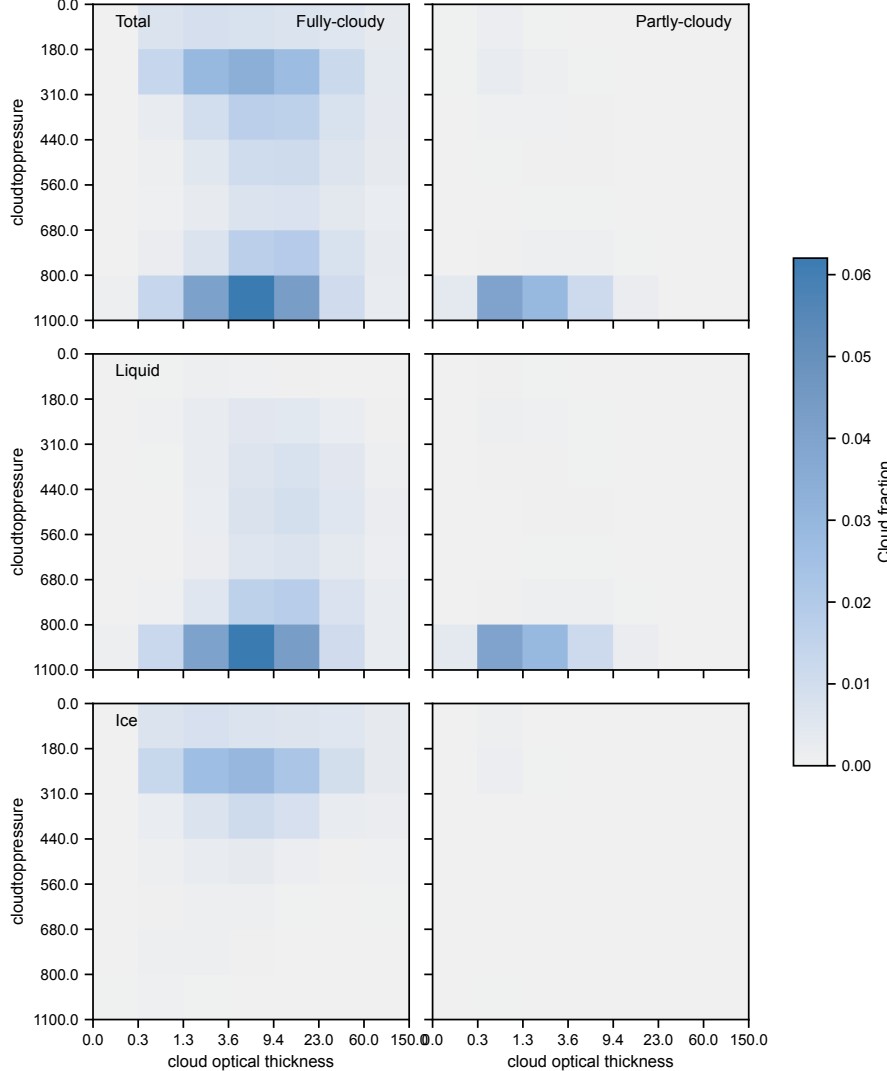

**Figure 7.** Joint frequency of cloud optical thickness ($\tau_c$, x-axis) and cloud top pressure $p_c$ (y-axis). Total/liquid/ice from top to bottom; fully- and partly-cloudy from left to right. July 2021.

spatial and temporal variations. The observations span the entire range of possible particle size values for both liquid and ice clouds regardless in both the fully- and partly-cloudy populations of pixels.

MODIS retrievals are formulated in terms of optical thickness and particle size (see Sec. 2.1) but the product of these quantities is used to estimate the cloud (liquid or ice) water path for each pixel. The MODIS-COSP datasets provides joint histograms of these values with particle size to support studies of aerosol-cloud interactions, where the ability to disentangle aerosol impacts on particle size from impacts on cloudiness and water path may be valuable. Figure 9 shows an example of



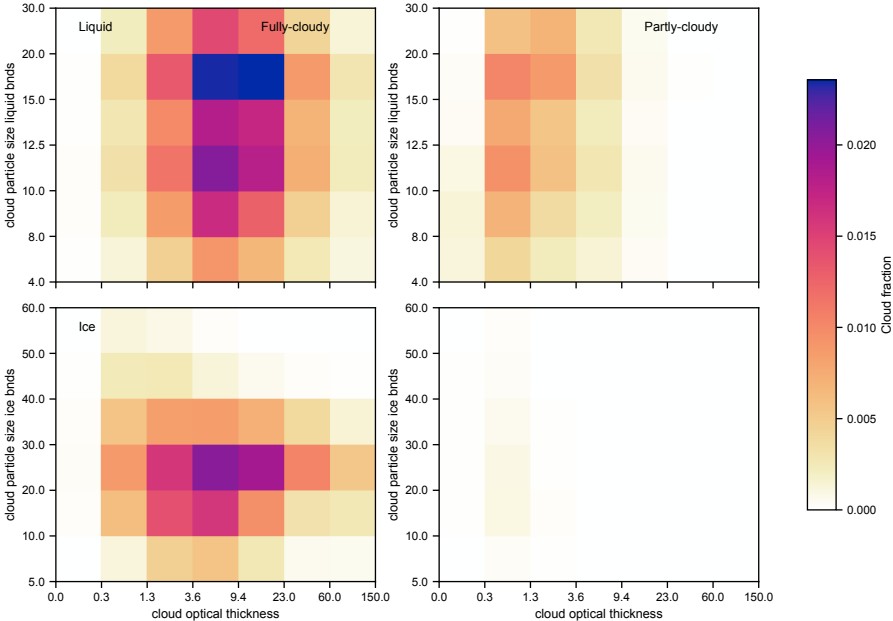

**Figure 8.** Joint frequency of cloud optical thickness ($\tau_c$, x-axis) and effective particle size $r_e$ (y-axis), shown as global (area-weighted) means for July 2021. Liquid clouds are in the top row and ice in the bottom row; fully cloudy pixels are in the left column and the much smaller number of partly-cloudy pixels in the right. Bins of optical thickness (x-axis) are spaced quasi-logarithmically; particle size bins are non-uniform in an attempt to resolve variations. Partly-cloudy pixels tend to be optically thin, especially for ice(see also Fig. 7), with particle sizes spanning the same range as the fully-cloudy pixels.

these histograms averaged over the globe for a single month. These histograms are not yet available from the MODIS simulator but we expect to add the capability to produce them in the coming months.

## 4   Comparing the data with the MODIS simulator and other MODIS datasets

### 4.1   Differences with respect to standard Level 3

The MODIS COSP Level-3 dataset MCD06COSP is provided as a convenience to users of COSP and the MODIS simulator. Differences relative to the standard MODIS Level-3 products, described throughout this paper, may be summarized as:

1. The dataset contains observations from both the Terra and Aqua platforms

2. Height-resolved cloud fractions from the cloud mask are reported

3. Cloud fractions for liquid and ice clouds, as determined by the cloud optical properties processing, are reported

4. Joint histograms of cloud optical thickness with effective radius are reported more compactly





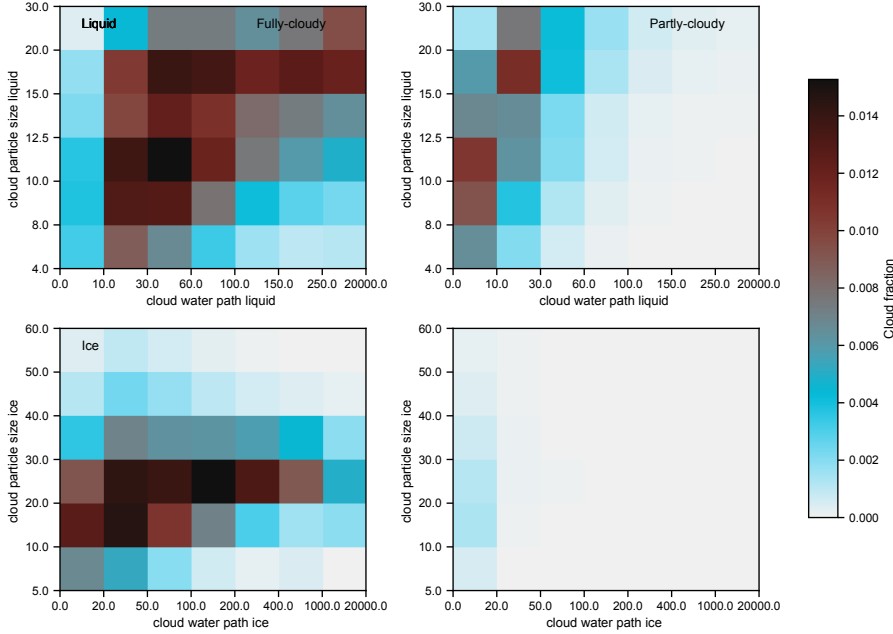

**Figure 9.** Joint frequency of cloud water path and cloud particle size, shown as global (area-weighted) means for July 2021, with thermodynamic phase in rows and fully-partly pixels separated by column.

5. Data are restricted to daytime observations

6. All scalars represent the underlying population of pixels, where temporal averages of cloud mask and cloud-top pressure datasets in the standard Level-3 products report weight each day equally

7. Data is provided in netCDF4 files

## 4.2 Comparisons to the MODIS simulator

The MODIS simulator, like COSP as a whole, operates on single atmospheric states; aggregation over time is the responsibility of the user. We stress that the MODIS-COSP data is designed to represent the underlying population of cloudy pixels observed in each region over the course of a day or month, so that "mean" values represent the in-cloud mean and not the domain mean. This implies that mean values from the MODIS simulator must be weighted in time by the cloud fraction to be directly comparable to the monthly data set reported here. Users might also choose to compute their own monthly mean observations, weighting each day equally, from the daily data.

To facilitate comparisons with the MODIS simulator we have provided Python code, described below, that transforms a set of monthly files containing all variables to datasets with time series of each variables, which may be written as netCDF files and/or Zarr stores.





## 5 Code and data availability

The data described in this paper are available for download from the NASA Level-1 and Atmosphere Archive & Distribution
System (LAADS) Distributed Active Archive Center (DAAC) in Greenbelt, Maryland, US. The daily data may be cited as
NASA (2022a, doi:10.5067/MODIS/MCD06COSP_D3_MODIS.062). The citation for the monthly data is NASA (2022b,
doi:10.5067/MODIS/MCD06COSP_M3_MODIS.062).

Code used to create the figures in this manuscript, including code for downloading and post-processing the data and making
the figures themselves, is available at https://github.com/RobertPincus/MODIS-COSP-data. A doi from Zenodo will be sought
at acceptance.

*Author contributions.* RP worked with PAH to define the data product and coordinated this paper. PAH implemented the processing stream
that produces the data described here and wrote the users' guide from which this paper is abstracted. SEP and KM are responsible for the
scientific production of MODIS cloud retrievals including the data described here. PAH, REH, and DB shepherded the production of data.
CJW motivated and helped to design the water path/particle size joint histograms.

*Competing interests.* The authors declare no competing interests.

*Acknowledgements.* We are grateful to Paulo Ceppi and Mark Zelinka for encouraging the production of joint histograms of cloud top
pressure and cloud optical thickness and providing feedback on preliminary versions of the dataset.





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



**Table 1.** Scalar variables available in MODIS COSP files. For each variable the mean, standard deviation, sum, sum-of-squares, and number of observations (pixel counts) is reported in each $1°$ latitude-longitude bin; the latter variables are primarily useful in aggregating over time (e.g. producing monthly files from daily files). Joint histograms are reported as integers and may be normalized by the corresponding number of observations to determine the fractional cloudiness. High clouds are those with cloud top pressure $p_c < 440\,\mathrm{hPa}$; mid-level clouds those with $680 < p_c \leq 440\,\mathrm{hPa}$; low clouds have $p_c \geq 440\,\mathrm{hPa}$. Total cloud fraction from the cloud mask is the sum of probably and confidently clear divided by the total of determined pixels. Total cloud fraction and cloud optical thickness from the cloud retrievals include clouds unknown phase, determined assuming liquid phase. With the exception of the geometric-mean optical thickness, all scalar measures of cloud properties include separate summaries for partly-cloudy pixels (i.e. Cloud_Retrieval_Fraction_Total and Cloud_Retrieval_Fraction_PCL_Total), so the total number of data groups is ten more than the 22 listed below.

| Parameter | Subsets | netCDF group name(s) |
|---|---|---|
| Viewing and illumination | Day Mask: SZA $\leq 85°$ | |
| Solar zenith angle (SZA) | | Solar_Zenith |
| Solar azimuth angle | | Solar_Azimuth |
| Sensor zenith angle | | Sensor_Zenith |
| Sensor azimuth angle | | Sensor_Azimuth |
| Derived from cloud top properties | Day Mask: SZA $\leq 85°$ | |
| Cloud top pressure $p_c$ | | Cloud_Top_Pressure |
| Cloud fraction – mask | total; high, middle, low | Cloud_Mask_Fraction |
| | | Cloud_Mask_Fraction_Low |
| | | Cloud_Mask_Fraction_Mid |
| | | Cloud_Mask_Fraction_High |
| Derived from cloud optical properties | Day mask: SZA $\leq 81.3731°$ | |
| Cloud fraction – retrieval | total; liquid, ice | Cloud_Retrieval_Fraction_Total |
| | | Cloud_Retrieval_Fraction_Liquid |
| | | Cloud_Retrieval_Fraction_Ice |
| Cloud optical thickness $\tau_c$ | total; liquid, ice | Cloud_Optical_Thickness_Total |
| | | Cloud_Optical_Thickness_Liquid |
| | | Cloud_Optical_Thickness_Ice |
| $\log_{10}(\tau_c)$ | liquid, ice, total | Cloud_Optical_Thickness_Log10_Liquid |
| | | Cloud_Optical_Thickness_Log10_Ice |
| | | Cloud_Optical_Thickness_Log10_Total |
| Cloud particle size | liquid, ice | Cloud_Particle_Size_Liquid |
| | | Cloud_Particle_Size_Ice |
| Cloud water path | liquid, ice | Cloud_Water_Path_Liquid |
| | | Cloud_Water_Path_Ice |



**Table 2.** Joint histograms available in MODIS COSP files. Joint histograms with particle size are provided for liquid and ice clouds separately; joint histograms with cloud top pressure are also available for all clouds. Joint histograms of partly-cloud pixels are accumulated separately (e.g. Cloud_Optical_Thickness_Total:JHisto_vs_Cloud_Top_Pressure and Cloud_Optical_Thickness_PCL_Total:JHisto_vs_Cloud_Top_Pressure) so that 14 joint histograms are available in total. See Figures 7 and 8.

| Joint frequency of occurence | netCDF group and variable names |
| --- | --- |
| Cloud optical thickness with cloud top pressure | Cloud_Optical_Thickness_Total:JHisto_vs_Cloud_Top_Pressure |
| | Cloud_Optical_Thickness_Liquid:JHisto_vs_Cloud_Top_Pressure |
| | Cloud_Optical_Thickness_Ice:JHisto_vs_Cloud_Top_Pressure |
| Cloud optical thickness with cloud particle size | Cloud_Optical_Thickness_Liquid:JHisto_vs_Cloud_Particle_Size_Liquid |
| | Cloud_Optical_Thickness_Ice:JHisto_vs_Cloud_Particle_Size_Ice |
| Cloud water path with cloud particle size | Cloud_Water_Path_Liquid:JHisto_vs_Cloud_Particle_Size_Liquid |
| | Cloud_Water_Path_Ice:JHisto_vs_Cloud_Particle_Size_Ice |