# Peer review of "Updated observations of clouds by MODIS for global model assessment"

_Earth System Science Data, 2022_

## Author Response (AR1)

Earth Syst. Sci. Data Discuss., author comment AC1
https://doi.org/10.5194/essd-2022-282-AC1, 2022
**Reply to reviewer comments on essd-2022-282**

Robert Pincus et al.
* * *
Author comment on "Updated observations of clouds by MODIS for global model assessment" by Robert Pincus et al., Earth Syst. Sci. Data Discuss., https://doi.org/10.5194/essd-2022-282-AC1, 2022
* * *
We are grateful for the prompt and constructive feedback provided by all three reviewers. We apologize for the length of time in responding which was partly the result of the pandemic. Below we describe how we have modified the manuscript to address the concerns. We have corrected typographic and graphical mistakes identified by the reviewers without further comment. Reviewer comments below are italicized.

**Common concerns**

Three comments by two reviewers focused on the clunkiness of the data set as provided:

Rev 1: *Lines 260-263: "To facilitate comparisons with the MODIS simulator we have provided Python code, described below, that transforms a set of monthly files containing all variables to datasets with time series of each variables, which may be written as netCDF files and/or Zarr stores." Does this sentence mean that the Python code takes individually saved netCDF files that each contain one month of data and simply concatenates the files into a time series for any individual variable and time period of choice? Please clarify.*

Rev 3: *I am a bit surprised that the dataset is not produced in such a way as to be applicable to models "right out of the box" as is the case for datasets like GOCCP. The motivation for leaving some processing as an exercise for the end-user, which seems ripe for accidental misuse, was not clear to me (though the provided python code is of course welcome). Is it an attempt to leave some flexibility for end-users' diverse needs?*

Rev 3: *I am also a bit surprised (as I am currently downloading the dataset via the github instructions) that the filenames have such cryptic names, particularly the timestamp which seems to be reporting a Julian day at the start of the month rather than a format like YYYYMM. This results in further reliance on the python script rather than being able to quickly assess what a file contains.*

We have expanded the discussion at the end of section 2 to explain why the data are provided in this less-than-ideal format, which is the result of the constraints under which the data are produced. We've moved the discussion of the simplifying Python scripts to the same location and elaborated on others tools we've made available.

Reviewers 2 and 3 found our choice of different color schemes to plot different-but-related quantities to be distracting:

*Figures: The authors clearly had a lot of fun trying out various matplotlib color schemes... I wonder if this may be distracting and unintentionally conveying differences that are not meant to be conveyed, considering many figures show the same field (cloud fraction). I am not sure whether any of these are unfriendly for color-blindness, but that should also be considered. The scheme in Figure 9 seems to artificially distinguish cloud fractions larger than about 0.007 from those below, but I'm not sure why that would be useful. In some figures lighter colors = larger values, but the opposite is true for Figures 3, 7-9.*

*Colour scales: please rationalise the use of the colour scales. Many different colour scales are used for no apparent reason. I suggest to consolidate all of them into two: dark to white, white to dark.*

We have experimented with using more uniform color scale, e.g. with using a grey scale in Figures 1-5. Recognizing that this is somewhat a matter of opinion we found this more confusing, since it lulls readers into the mistaken sense that they are comparing the same quantity. We have now noted in both text and figure captions that each distinct physical quantity is plotted with a unique color scale. We have ensured that brighter colors represent more and/or more reflective clouds in the maps (Figures 2, 4, 5) and have noted in the caption to figure 3 that darker colors indicate more dramatic differences. The color scales are from the "Colorcet" package from Holoviz and are indeed designed with various color-blindnesses in mind. The use of darker colors to indicate larger values in joint histograms follows the conventions used in plotting such histograms (e.g. doi:10.1175/JCLI-D-11-00248.1).

**Reviewer 1**

*The authors state that the product is made for the "convenience" of end users, but also mention that on line 39: "The system was also quite fragile and ceased production when NASA updated the production of MODIS datasets" in the Introduction. It's not transparent to me what is meant by that the system was quite fragile and why production ceased. Also the statement on line 46: "The dataset, produced using a system designed to be more robust to changes in the upstream data, provides a set of custom cloud-related parameters using specific dataset definitions more closely aligned with the MODIS simulator than are the standard datasets" is also not transparent to me. What upstream data are the authors referring to and what changes were made to them? Also, importantly, does this mean that using the standard MODIS product to compare against climate models is incorrect or is this product really only designed for the convenience of end users? Please clarify.'*

We have revised the introduction to make our goals more explicit although we hesitate to spend much time describing datasets that are no longer being produced. We have enumerated a longer list of barriers, added language to emphasize that our data set is a technical convenience, and been more explicit about "upstream data." We have now emphasized that the present data is a direct aggregation of the pixel-scale observations (the older data were not) but have not explained why the older data production ceased.

*A number of variables are described but I would recommend that in each case the variables themselves (as listed in Table 1) are spelled out to be clear and equations written out if applicable, which would be relevant for a publication in ESSD, e.g. line 257: when weighting by the "cloud fraction" in the MODIS simulator, Section 3.3 variables.*

That the observations are averaged to reflect the underling population, rather than assuming that each day is equally well-sampled, is explained three times in the

manuscript. We address the time averaging of simulations in an expanded section 4.2 described more fully in the response to reviewer 3.

*Why is there no height-resolved cloud retrieval fraction saved as a separate product? Only the height-resolved cloud fraction from the cloud mask seems to be reported.*

As we noted on line 145 " Summing the [joint histogram of cloud optical thickness and cloud top pressure] over all optical thickness bins and reducing the resolution in cloud-top pressure allows users to compute high, middle, and low cloud fractions consistent with cloud optical properties (as opposed to the cloud mask)."

*Is there a reason why this manuscript was submitted when the complete dataset is still under development? It seems that it would make more sense for the product to be finished first and then a final paper to be published on it for completeness and to avoid confusion in the future where the same product might be separately documented. It seems that the current User's Guide that is already available on the Internet is sufficient in the interim?*

We are unclear what the reviewer means here. The data set is complete though we are still advocating for continuous updating. The user's guide is certainly valuable but it is extremely detailed, mutable, and has not been subject to peer review.

*Figure 8: So if a user wanted the total joint histogram for liquid and ice, could they simply sum up the liquid and ice separately?*

They could. As noted at line 85, this would exclude pixels for which the phase could not be identified, though these pixels would contribute to the "Total" or unsegregated histogram.

*Lines 136-137: "…and the condensed water path estimated from the product of optical thickness and particle size". For ice clouds, what equation was used?*

To keep the focus on the aggregated data set we are producing we refer readers to the papers describing the MODIS pixel-scale data.

**Reviewer 2**

*L69-78: Is the 5km resolution true or nominal? It is not clear to me if the number of pixels that go into the calculation of the cloud fraction depends on the VZA. The text seems to imply that 25 '1km pixels' are used for all VZAs.*

Thanks; we've clarified this in the text, and indeed the reviewer is correct.

*L139-142: it would be helpful to explain why this approach for computing monthly averages is used.*

We have added a phrase explaining that the time-averaging used by standard MODIS products makes the tacit assumption that each day is equally well-sampled.

*L183-184: please can you quantify the impact of this bug, how small is "small"?*

As the data are missing it's hard to gauge the true impact. We now comment on the lower limit: cloud fraction is based on fewer pixels than is cloud mask in less than 1% of monthly-mean grid cells.

*Tables 1 and 2: Is it possible to report the name of the equivalent field from the MODIS simulator in a 4th column, or state that it is not available?*

We have instead added text at the end of the table captions to guide users. Adding this column makes the table overflow the page.

**Reviewer 3**

*If the COSP run in climate models produces monthly mean cloud property fields (cloud fraction, joint histograms, log(tau), etc), why can't this dataset provide the same right out of the box? Lines 257-260 indicate that even the model output will have to be further processed in order to match what is provided here, another place where user error can creep in (which fields require weighting by cloud fraction? Does cloud fraction have to be weighted by cloud fraction? Which cloud fraction -- mask or retrieval -- should be used as the weight?). Is there a plan to provide python code for processing the MODIS simulator output in such a way as to be directly comparable to what is produced by the provided python code? Could all of this post-processing be avoided from the outset by just providing an idiot-proofed dataset that is as close as possible to MODIS simulator output?*

*For several fields, it is not clear to me that there is a COSP counterpart; what is the reasoning for including these fields in the "MODIS-COSP" dataset if COSP does not provide them? These fields include the two versions of cloud fraction (from the mask and from the retrieval); the partly cloudy pixel fields; and the additional statistics like standard deviations, sum-of-squares, etc.*

Both of these perceptive questions focus on the ability to compare the observational data described here to output from the MODIS simulator. We have revised and added to section 4.2 describing these comparisons. On the technical front we note that time aggregation is normally done within climate models as the simulation advances. We highlight the observational averaging strategy specifically so that users can implement time aggregation correctly. (The post-processing solution proposed by the reviewers won't usually apply.) We have added material emphasizing how definitions of cloud fraction differ between observations and the proxy and pointing readers to previous work exploring how these might be reconciled.

---

## Author Response (AR2)

**Response to reviewers**

We thanks reviewer 3 for his or her very careful and close reading of our manuscript and apologize the oversights and errors. In most cases we have adopted the reviewers corrections without comment. A few changes or lack thereof are worth noting:

*Figure 2 caption: the last sentence does not belong here. This is stated in the main text, where it is more appropriate.*

We acknowledge that the caption repeats material in the text; that's a deliberate attempt to get the most important points to less careful readers.

*L248-249: is there a reference showing that this product equals water path? It is not immediately obvious to me that multiplying these two things would yield a water path.*

Stephens (1978) noted that optical depth equals 3/2 times the liquid water path divided by the effective radius under certain assumptions; this result is widely known. Since our point is only the water path is derived from optical depth and effective particle size, rather than representing an independent measurement, we have added modified the text to add "appropriately-scaled product" and a reference rather than providing a more detailed explanation.

*-L274: Does the MODIS simulator also mimic the sampling of the Aqua/Terra satellites, or does it sample every geographic location at every timestep?*

We have added a phrase to highlight that orbital sampling is the responsibility of the host model.

*L277-279: I'm still confused about what your recommendation is here, and for whom it is directed. Does this mean that an end-user cannot compare this dataset directly to monthly mean COSP output available from, e.g., CMIP, and that one has to create monthly means from daily COSP output via the same aggregation strategy employed herein? Or is this a message to model developers implementing COSP in their models to ensure that the COSP fields are aggregated properly to produce monthly means in accord with this product? Please clarify.*

We have revised the text to emphasize that the time averaging to match these observation is the responsibility of model developers.